# TopoLogic: An Interpretable Pipeline for Lane Topology Reasoning on Driving Scenes

**Yanping Fu**
Institute of Computing Technology,
Chinese Academy of Sciences
University of Chinese Academy of Sciences
fuyanping23s@ict.ac.cn

**Wenbin Liao**
Institute of Computing Technology,
Chinese Academy of Sciences
University of Chinese Academy of Sciences
liaowenbin23z@ict.ac.cn

**Xinyuan Liu**
Institute of Computing Technology,
Chinese Academy of Sciences
University of Chinese Academy of Sciences
liuxinyuan21s@ict.ac.cn

**Hang Xu**
Hangzhou Dianzi University
hxu@hdu.edu.cn

**Yike Ma**
Institute of Computing Technology,
Chinese Academy of Sciences
ykma@ict.ac.cn

**Yucheng Zhang**
Institute of Computing Technology,
Chinese Academy of Sciences
zhangyucheng@ict.ac.cn

**Feng Dai**[*]
Institute of Computing Technology,
Chinese Academy of Sciences
fdai@ict.ac.cn

## Abstract

As an emerging task that integrates perception and reasoning, topology reasoning in autonomous driving scenes has recently garnered widespread attention. However, existing works often emphasizes "perception over reasoning": they typically boost reasoning performance by enhancing the perception of lanes and directly adopt vanilla MLP to learn lane topology from lane query. This paradigm overlooks the geometric features intrinsic to the lanes themselves and is prone to being influenced by inherent endpoint shifts in lane detection. To tackle this issue, we propose an interpretable method for lane topology reasoning based on lane geometric distance and lane query similarity, named TopoLogic. This method mitigates the impact of endpoint shifts in geometric space, and introduces explicit similarity calculation in semantic space as a complement. By integrating results from both spaces, our method provides more comprehensive information for lane topology. Ultimately, our approach significantly outperforms the existing state-of-the-art methods on the mainstream benchmark OpenLane-V2 (23.9 v.s. 10.9 in TOP$_{ll}$ and 44.1 v.s. 39.8 in OLS on *subset_A*). Additionally, our proposed geometric distance topology reasoning method can be incorporated into well-trained models without re-training, significantly boosting the performance of lane topology reasoning. The code is released at https://github.com/Franpin/TopoLogic.

---

[*]Corresponding Author

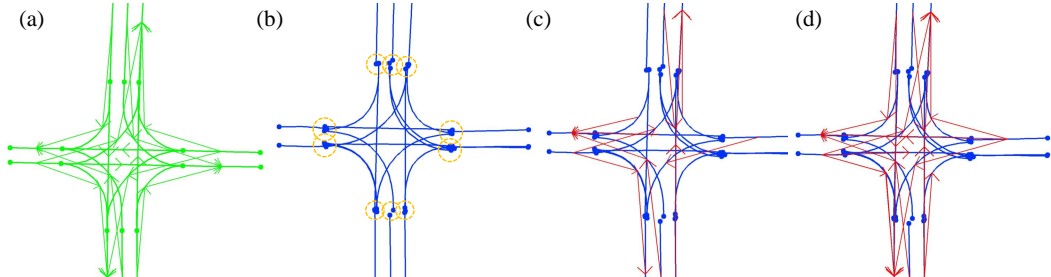

Figure 1: **Comparison of results with and without post-processing in TopoNet.** We use a post-processing based on geometric distance to improve the lane topology reasoning performance of TopoNet. **(a)** denotes the ground truth of lane topology reasoning. **(b)** denotes the endpoints of two connected lanes in prediction do not overlap (marked with yellow circle) as desired in ground truth. **(c)** denotes the lane topology reasoning result of TopoNet, the arrow denotes lane topology (marked with red arrow). **(d)** denotes the lane topology reasoning result of TopoNet using post-processing, significantly improves the reasoning precision of lane topology.

# 1   Introduction

In recent years, the field of autonomous driving has witnessed numerous milestone achievements and has progressively shifted from pure research to practical applications. In complex driving scenarios, vehicles need to perceive lanes & traffic elements and reason their topological relationships (i.e., lane connectivity and correspondence with traffic elements), which provides comprehensive information for downstream path planning and motion control. Under the trend of end-to-end autonomous driving, abovementioned perception and reasoning are integrated into a single task, referred to as topology reasoning [1] in autonomous driving scenes. This challenge has attracted widespread attention within the ego planning [2, 3, 4]and high-definition map learning [5, 6, 7, 8] communities.

The topology reasoning task has garnered significant attention recently, since it is closer to the real needs. Some works have explored lane centerline representation [1, 9, 10, 11] and lane segment representation [12], while others have introduced SDMap (Standard-Definition Map) [13] to provide additional clues for learning. However, existing works primarily focus on enhancements in the perception part, with scarce modifications made to the reasoning part. Irrespective of the approach details, existing studies typical employ vanilla MLP to learn lane topology directly from lane query. This paradigm has its shortcoming: since each lane is encoded independently through distinct query, it is challenging to ensure strictly overlap at the endpoints of two connected lanes, as shown in Figure 1(b). In contrast, it is evident that the endpoints of two connected lanes in the ground truth actually overlap perfectly, as shown in Figure 1(a). Lanes with slightly shifted endpoints may be erroneously classified by MLP as disconnected. This leads to MLP easily predicting fewer lane topology, as shown in Figure 1(c).

To tackle the aforementioned issues, we introduce TopoLogic, an interpretable method for lane topology reasoning that is based on lane geometric distances and the similarity of lane query in semantic space. The geometric distance-based approach aims to mitigates the impact of endpoint shift, thereby more robustly learning lane topology. This approach first calculates the geometric distance between lanes and then uses a learnable mapping function to map the distance to connectivity probability. Notably, for two given lanes, their geometric distance is defined as the distance between the ending point of one lane and the starting point of subsequent lane. This distance itself can serve as a strong criterion: when this distance is within a certain range, the predicted endpoints should be considered overlapping, and the lanes connected; otherwise, they are not. In this way, the lane topology reasoning becomes more tolerant of endpoint shifts, thus becoming more accurate. It's worth noting that even when the geometric distance method is merely applied as post-processing without re-training, the performance of SOTA model in lane topology reasoning is significantly improved, as shown in Figure 1(d). Moreover, reasoning lane topology completely based on geometric distances can lead to inaccuracies when lane detection is imprecise as is shown in Figure 4, since the calculation of lane geometric distance heavily relies on the accuracy of lane detection. To make up for the deficiency of geometric distances approach, we design a extra topology approach based on the similarity of lane

queries as a complement. This approach projects lane queries into a high-dimensional semantic space, and involves explicitly computing the dot product between lane query to determine similarity, and then mapping this similarity onto lane topology using sigmoid [14]. The approach for calculating lane query similarity complements the approach used for computing lane geometric distance topology and similarly boasts high interpretability. The final lane topology is obtained by fusing the topology matrix derived from both approachs. By the way, the lane topology is also used in GNN to enhance lane learning through the aggregation of features from adjacent lanes. **In summary, our contributions are as follows:**

**1.** We identify the current state of research in topology reasoning as "perception over reasoning", and reveal that the lane topology is easily disturbed by the endpoint shifts in lane detection when MLP employed solely for lane topology reasoning.

**2.** We propose an interpretable method, referred to as TopoLogic, which conducts lane topology reasoning by calculating lane geometric distances and semantic similarity of lane query in a high-dimensional semantic space.

**3.** Extensive experiments on the mainstream benchmark OpenLane-V2 for topology reasoning task indicate that our method significantly outperforms existing state-of-the-art methods, especially in lane topology metric. Even if employed solely as a post-processing step without re-training, proposed geometric distance approach can significantly enhance well-trained lane topology reasoning models.

## 2    Related Work

### 2.1    Lane Detection

Lane detection plays an important role in autonomous driving, which has been a fundamental aspect of lane topology reasoning. In the realm of lane detection, some works [15, 16, 17] attempt to perform lane detection on a segmentation map. Moreover, some researchers use vector-based methods to perform 3D lane detection [18, 19, 20, 21], however, these methods rely on a predetermined series of Y-axis coordinates within the query for forecasting 3D lanes, thereby lacking the capability to exclusively predict 3D lane positions along the Y-axis. In recent study, TopoNet [1] leverages Graph Neural Network (GNN) [22] to enhance the perception of lane centerline, while TopoMLP [9] utilizes PETR [23] for centerline detection. LaneSegNet [12] designs a Lane Attention mechanism to reinforce the perception of lane segment, and SMERF [13] introduces Standard-definition (SD) Map as an additional input to bolster the perception of lane centerline. In our work, we enhance lane learning by aggregating features of adjacent lanes through GNN, which involves computing lane geometric distance and lane query similarity.

### 2.2    Lane Topology Reasoning

In lane topology reasoning, accurate comprehension of lane topology is imperative for effective navigation and decision-making in autonomous driving. Some methods [24, 25, 26, 27, 28] have been proposed to address this. The STSU [29] model drew inspiration from DETR [30] and employed a neural network architecture, complemented by a multi-layer perceptron (MLP) to establish line connectivity. Building upon this foundation, Can et al. [31] introduced minimal cycle queries to refine centerline, ensuring accurate ordering of overlapping lines and thereby enhancing precision. Further advancements include the perception of centerline [1, 13, 11, 9] and the perception of lane segment [12]. Among them, CenterLineDet [11] and TopoNet[1], Both treat lane line as vertices and leverage an graph-based model to update lane representation and lane topology. While these methods have predominantly relied on MLP for generating adjacency matrices to represent lane topology. In our work, we calculate the lane topology matrices based on the geometric distances between lanes and the similarity of lane query within high-dimensional semantic space, respectively, and then fuse them to form the final lane topology. The fusion of geometric and semantic space enriches the model's understanding of lane topology, thereby culminating in improved performance in driving scene analysis and decision-making.

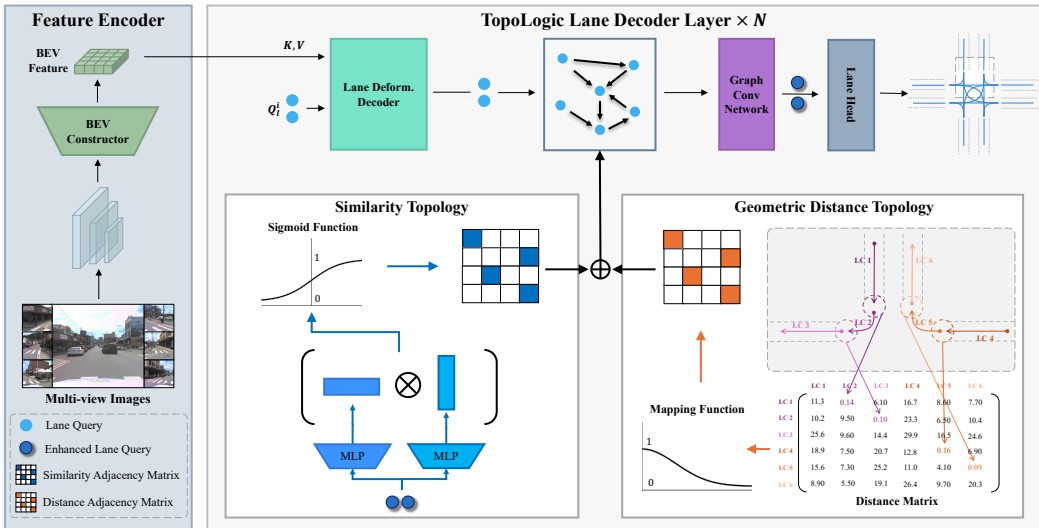

Figure 2: **Pipeline of TopoLogic.** The overarching structure of TopoLogic comprises two main components: an image encoder for feature extraction and transformation, and a lane decoder responsible for end-to-end topology reasoning. This decoder utilizes the proposed lane geometric distance topology and lane similarity topology, and fuse them into the final lane topology, which is facilitated through GNN to augment lane learning in the next decoder layer.

## 3 Method

### 3.1 Problem Definition

Given images captured by the surround-view cameras of a vehicle, lane topology reasoning needs to perceive lane instances in BEV(Bird's Eye View) and then infer the topology between these lane instances. The enhancement of lane instance perception assists in the reasoning of lane topology. Lane instances are described as a set of directed lane lines which is denoted as $L = [l_0, \ldots, l_{n-1}]$. Each lane line is composed of a series of ordered points, and it is denoted as $l = [p_0, \ldots, p_{n-1}], p = (x, y, z) \in \mathbb{R}^3$. The topology between lane instances signifies the connectivity of the directed lanes and it is depicted as a topology graph $(V, E)$, where the edge set $E \subseteq V \times V$. An entry $(i, j)$ in $E$ is positive if and only if the ending point of lane $l_i^{end}$ connects to the starting point of lane $l_j^{start}$.

### 3.2 Overview

As is depicted in the Figure 2, our proposed TopoLogic takes multi-view images from onboard cameras as input. These images are processed by a backbone to generate multi-scale image features. Multi-scale image features are transformed into BEV features through a view transformation module, and then passed to a lane deformable decoder to generate lane query $Q_l$ for lane detection. The proposed lane geometric distance approach and the lane similarity approach compute the lane topology respectively. Ultimately, the two topologies are fused and fed into GNN to augment the learning of lane line in the next decoder layer.

### 3.3 Lane Geometric Distance Topology

**Lane Geometric Distance Matrix.** Lane query $Q_l$ can generate multiple directed lane lines through lane head. We can assess the connectivity between these lanes by computing the geometric distance between the ending point of one directed lane line and the starting point of the following lane line.

$$l_0, \ldots, l_{n-1} = \text{LaneHead}(Q_l^i) \tag{1}$$

$$d_{ij} = \mid l_i^{end} - l_j^{start} \mid \tag{2}$$

$$D = \{d_{ij} \mid i, j = 0 \ldots n-1\} \tag{3}$$

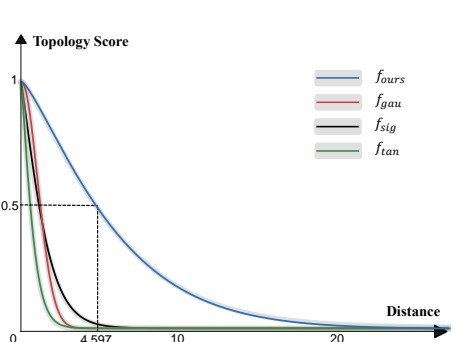

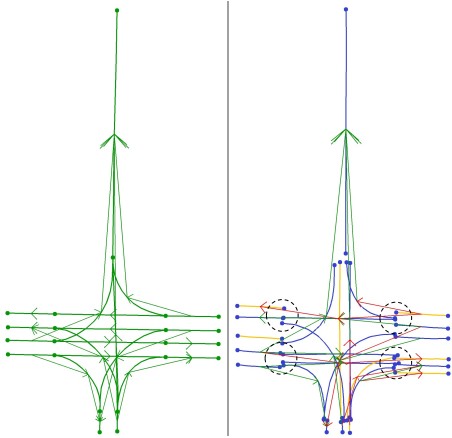

Figure 3: **Comparison of various mapping functions.** $f_{gau}$ represent Gaussian function, $f_{sig}$ represent sigmoid function, and $f_{tan}$ represent tanh function. Compared to $f_{gau}, f_{sig}, f_{tan}$, our proposed function $f_{ours}$ has greater tolerance for endpoint shift.

Figure 4: **Influence of inaccuracies in lane detection.** Blue denotes correct prediction of lane line, yellow denotes incorrect prediction of lane line, green denotes correct prediction of lane topology, and red denotes incorrect prediction of lane topology.

where $D$ is the lane geometric distance, $l_i$ and $l_j$ represent preceding and subsequent lane lines, $d_{ij}$ denotes the geometric distance between $l_i$ and $l_j$, $l_i^{end}$ signifies the last point of lane line $l_i$, and $l_j^{start}$ indicates the first point of lane line $l_j$.

**Distance to Topology Mapping Function.** After obtaining the geometric distance matrix $D$ for the lanes, it is necessary to map the lane geometric distance into the lane topology. The lane topology can be represented by a matrix ranged in 0~1. Zero indicates that there is no connection between two lanes, while one indicates that there is a connection. This mapping function needs to capture the following notion: when the input $x \to 0$, it meaning the two lanes are very close to each other, the output $y \to 1$, suggesting that these two lanes are very likely to be connected. Conversely, as $x \to \infty$, $y \to 0$. Inspired by the Gaussian function, we design a learnable mapping function as follows:

$$f_{ours} = e^{-\frac{x^{\alpha}}{\lambda \cdot \sigma}} \tag{4}$$

where $x = d_{ij}$. $\sigma$ is the standard deviation of the geometric distance matrix $D$. $\alpha, \lambda$ are learnable parameters. With the help of such mapping, we can get a lane topology as follows:

$$G_{dis} = \{f_{ours}(d_{ij}) | i, j = 0...n - 1\} \tag{5}$$

There also exists some common alternative functions that meet the criteria, for example Gaussian function, sigmoid-based function and tanh-based function as Equation 6(a,b,c). We make a comparison between them with $f_{ours}$ in Figure 3. Obviously, $f_{ours}$ sets a larger geometric distance threshold for determining topological connectivity compared to $f_{gau}$, $f_{sig}$ and $f_{tan}$, which makes the lane topology more robust to the endpoint shifts. Ablation study in Table 3 also verifies this opinion.

$$f_{gau} = e^{-\frac{x^2}{2}} \quad (a), \quad f_{sig} = \frac{2}{1 + e^x} \quad (b), \quad f_{tan} = \frac{e^{-x} - e^x}{e^{-x} + e^x} + 1 \quad (c) \tag{6}$$

### 3.4 Lane Similarity Topology

Lane topology reasoning based on the geometric distance of lane lines can achieve commendable results when the detection of lane lines is accurate. However, since this topology reasoning method heavily relies on the detected lane lines, inaccuracies in lane line detection can interfere with the geometric approach and lead to erroneous reasoning outcomes, as demonstrated in the Figure 4. In light of this situation, we reason lane topology by calculating the similarity between lane query $Q_l$ within high-dimensional semantic space. A higher similarity between $Q_l$ indicates a greater likelihood of connectivity between the lanes, while lower similarity suggests an absence of connectivity. We

initially encode $Q_l$ using two distinct MLP, and then represent the similarity by computing the inner product between the two encoding results. Finally, we require a function to map the similarity between $Q_l$ to lane topology. Given the correlation between lane similarity and lane topology, we employ sigmoid to map the lane similarity onto lane topology. This process is as follows:

$$Q_{emb_1}, Q_{emb_2} = \text{MLP}_1(Q_l^i), \text{MLP}_2(Q_l^i) \tag{7}$$

$$S = \text{matmul}(Q_{emb_1}, \text{transpose}(Q_{emb_2})) \tag{8}$$

$$G_{sim} = \text{sigmoid}(S) \tag{9}$$

where $Q_l^i \in \mathbb{R}^{N_l \times C}$, $S$ represents the similarity of $Q_l$. $G_{sim} \in \mathbb{R}^{N_l \times N_l}$, while $N_l$ represents the number of lane query.

### 3.5 Lane-Lane Topology

Both the lane topology reasoned from the geometric distance between lane lines and the lane topology reasoned from the similarity of lane query $Q_l$ in the high-dimensional semantic space can indicate the connectivity of lanes. These two methods are complementary in the task of lane topology reasoning. In this context, we merge the two lane topology reasoning results together as the final and more accurate lane topology using learnable coefficients as follow:

$$G = \lambda_1 \cdot G_{dis} + \lambda_2 \cdot G_{sim} \tag{10}$$

where $\lambda_1, \lambda_2$ are learnable parameters, and $G$ is the final lane topology prediction.

### 3.6 Learning

Similar to Transfomer-based networks [30, 32], the supervision is applied on each decoder layer to optimize the query feature iteratively. The overall loss of TopoLogic is:

$$\mathcal{L} = \mathcal{L}_{det} + \mathcal{L}_{top} \tag{11}$$

The lane detection loss $\mathcal{L}_{det}$ consists of a focal loss [33] for lane classification and an L1 loss for lane regression. The lane topology reasoning loss $\mathcal{L}_{top}$ includes only the loss computed for $G_{\text{sim}}$. As for the calculation of $G_{\text{dis}}$, since it is enhanced by GNN to facilitate lane learning, we update its learnable parameters through $\mathcal{L}_{det}$.

## 4 Experiment

### 4.1 Dataset and Metric

**Dataset**. We have evaluated TopoLogic on the OpenLane-V2 [34] , which is currently the only large-scale perception and topology reasoning dataset devised for autonomous driving scenarios. This dataset was developed by Argogorse2 [35] and nuScenes [36] respectively. It provides annotations for both lane centerline tasks and lane segment detection tasks. OpenLane-V2 consists of two subsets: *subset_A* and *subset_B*, each comprising 1000 scenes with 2Hz multi-view images and annotations. Each subset includes annotations for the lane centerline, traffic element, lane topology, as well as the topology between traffic element and lane. In *subset_A*, there are seven views as input, together with an additional Standard-definition Map input, and the annotations for lane segment have been expanded; *subset_B* contain only six views as input.

**Metric**. OpenLane-V2 evaluates perception tasks for both lane centerline and lane segment.

**(1)** In the task of lane centerline perception, the metrics include $\text{DET}_l$, $\text{DET}_t$, $\text{TOP}_{ll}$, and $\text{TOP}_{lt}$. $\text{DET}_l$ quantifies similarity by averaging the Frechet distance at matching thresholds of 1.0, 2.0, 3.0. $\text{DET}_t$ uses Intersection over Union (IoU) as a measure of similarity and calculates the average across different traffic categories. $\text{TOP}_{ll}$ and $\text{TOP}_{lt}$ respectively compute the topology matrix similarity between lanes and between lanes and traffic elements, with the overall evaluation metric for lane centerlines being denoted as OLS. The OLS is carried out as $\text{OLS} = \frac{1}{4}[\text{DET}_l + \text{DET}_t + f(\text{TOP}_{ll}) + f(\text{TOP}_{lt})]$, where $f$ is the square root function.

**(2)** In the task of lane segment perception, we adopt the metric proposed by LaneSegNet for evaluating lane segment perception. These include the lane segment distance $\text{D}_{ls}$, the corresponding Average

Precision $AP_{ls}$ and $AP_{ped}$, with the mAP calculated as the average of $AP_{ls}$ and $AP_{ped}$. The lane segment topology metric is denoted as $TOP_{lsls}$.

**(3)** For centerline, OpenLane-V2 has two versions available for evaluation on $TOP_{ll}$, $TOP_{lt}$, and OLS: v1.0.0 and v2.1.0. For lane segment, OpenLane-V2 has versions v2.0.0 and v2.1.0 on $TOP_{lsls}$.
[2] **Since the ultimate goal of perception is reasoning, we believe that topology metrics are what should be paid more attention. Moreover, our modifications mainly involve lane topology reasoning, so we primarily focus on the lane topology metric $TOP_{ll}$ and $TOP_{lsls}$.**

## 4.2 Implementation Details

**Feature Extractor.** All images are resized into the same resolution of 1550 × 2048. For reproducibility, we utilize the official implementations of TopoNet, SMERF, and LaneSegNet models. Both models use ResNet-50 [37] backbone pretrained on ImageNet [38] paired with a Feature Pyramid Network (FPN) [39] to extract multi-scale features. The number of output channels is set to 256. We employ the view transformer from BEVformer [40] encoder to transform multi-scale features into BEV features. The size of BEV grids is set to 200×100. TopoLogic is configured identically.

**Lane Detector.** We employ Deformable DETR [32] for the detection of lane line. The number of query is set to 200. After passing through each decoder layer of Deformable DETR, the query undergo GNN using the lane topology matrix. We predict the offset of the lane lines by setting reference points, with each lane line consisting of 11 three-dimensional points. For LaneHead, the classification head adopts a three-layer MLP with LayerNorm and ReLU to predict the confidence score of the lane line. The regression head is a three-layer MLP combined with ReLU, used to predict the 11×3 offset of the lane line. For lane detection loss $\mathcal{L}_{det_l}$, the weight of the classification part is 1.5, and the weight of the regression part is 0.025.

**Lane Topology Head.** The Lane Topology Head consists of a lane geometry distance predictor and a lane similarity predictor. For lane geometry distance predictor, we first calculate the geometric distance between the terminating point of the previous lane line and the starting point of the subsequent lane line to obtain a 200×200 distance matrix. Then the distance matrix is mapped to a lane topology matrix through a learnable mapping function $f(x) = e^{-\frac{x^\alpha}{\lambda \cdot \sigma}}$, where the size of $\alpha, \lambda, \sigma$ is 1×1, $\sigma$ is the standard deviation of $x$, $\alpha, \lambda$ are learnable parameters, $\alpha$ is initialized to 2, $\lambda$ is initialized to 0.2. For the calculation of lane similarity, given a 200×256 lane query, it is encoded through two different three-layer MLP. The we compute the similarity between the encoded results, resulting in a 200×200 lane similarity matrix. The similarity matrix is transformed into a lane topology matrix through sigmoid. The two lane topology matrices are fused into the final lane topology using learnable coefficients, which are initialized to 1 and have a size of 1×1.

**Training.** We train TopoLogic utilizing the AdamW optimizer [43] with a weight decay of 0.01 with an initial learning rate of $2 \times 10^{-4}$ and employ a cosine annealing schedule for the learning rate. All experiment is trained for 24 epochs on 8 NVIDIA RTX 3090 GPUs with a batch size of 16.

## 4.3 Comparison to State-of-the-art

**Centerline.** We compared TopoLogic with existing state-of-the-art methods such as STSU, VectorMapNet, MapTR, TopoNet, SMERF on centerline. Table 1 shows the results on the *subset_A* and *subset_B* datasets. Without any additions, our method achieves state-of-the-art performance. Compared with TopoNet, our method achieves decent detection accuracy (**29.9** v.s. 28.6 on *subset_A*, **25.9** v.s. 24.4 on *subset_B*), especially with a significant improvement in $TOP_{ll}$ (**18.6** v.s. 4.1 on *subset_A* for v1.0.0, **23.9** v.s. 10.8 on *subset_A* for v2.1.0), which is the lane topology reasoning score. There is also an improvement in OpenLane-V2 overall score OLS (**41.6** v.s. 35.6 on *subset_A* for v1.0.0, **44.1** v.s. 39.8 on *subset_A* for v2.1.0). Even when using SDMap, our proposed TopoLogic still manages to achieve state-of-the-art performance and realizes a significant improvement in $TOP_{ll}$ (**23.4** v.s. 7.5 for v1.0.0, **28.9** v.s. 15.4 for v2.1.0).

**Lane Segment.** Concurrently, we compared TopoLogic with existing state-of-the-art methods such as TopoNet, MapTR, MapTRv2, LaneSegNet on lane segment. In Table 2, it indicates that our method exhibits improvements in the Mean Average Precision (mAP) for lane segment detection compared

---

[2]See the official repository for version differences of metrics: `https://github.com/OpenDriveLab/OpenLane-V2/issues/76d`

Table 1: Performance comparison with state-of-the-art methods on OpenLane-V2 benchmark on centerline. Results for existing methods are from TopoNet, TopoMLP and SMERF. "SDMap" indicates the use of a Standard-definition Map. "-" denotes the absence of relevant data. We are more focused on $TOP_{ll}$.

| Data | Method | SDMap | $DET_l$ ↑ | $DET_t$ ↑ | v1.0.0 | | | v2.1.0 | | |
|------|--------|-------|-----------|-----------|--------|--------|------|--------|--------|------|
| | | | | | $TOP_{ll}$ ↑ | $TOP_{lt}$ ↑ | OLS↑ | $TOP_{ll}$ ↑ | $TOP_{lt}$ ↑ | OLS↑ |
| subset_A | STSU [16] | × | 12.7 | 43.0 | 0.5 | 15.1 | 25.4 | 2.9 | 19.8 | 29.3 |
| | VectorMapNet [6] | × | 11.1 | 41.7 | 0.4 | 5.9 | 20.8 | 2.7 | 9.2 | 24.9 |
| | MapTR [41] | × | 17.7 | 43.5 | 1.1 | 10.4 | 26.0 | 5.9 | 15.1 | 31.0 |
| | TopoNet [1] | × | 28.6 | 48.6 | 4.1 | 20.8 | 35.6 | 10.9 | 23.8 | 39.8 |
| | TopoMLP [9] | × | 28.3 | **50.0** | 7.2 | 22.8 | 38.2 | 19.0 | 23.4 | 42.2 |
| | **TopoLogic** | × | **29.9** | 47.2 | **18.6** | 21.5 | 41.6 | 23.9 | 25.4 | **44.1** |
| | SMERF [13] | ✓ | 33.4 | **48.6** | 7.5 | 23.4 | 39.4 | 15.4 | 25.4 | 42.9 |
| | **TopoLogic** | ✓ | **34.4** | 48.3 | **23.4** | 24.4 | 45.1 | 28.9 | 28.7 | 47.5 |
| subset_B | STSU [16] | × | 8.2 | 43.9 | 0.0 | 9.4 | 21.2 | - | - | - |
| | VectorMapNet [6] | × | 3.5 | 49.1 | 0.0 | 1.4 | 16.3 | - | - | - |
| | MapTR [41] | × | 15.2 | 54.0 | 0.5 | 6.1 | 25.2 | - | - | - |
| | TopoNet [1] | × | 24.3 | 55.0 | 2.5 | 14.2 | 33.2 | 6.7 | 16.7 | 36.8 |
| | TopoMLP [9] | × | **26.6** | **58.3** | 7.6 | **17.8** | 38.7 | - | - | - |
| | **TopoLogic** | × | 25.9 | 54.7 | **15.1** | 15.1 | 39.6 | 21.6 | 17.9 | 42.3 |

Table 2: Performance comparison with state-of-the-art methods on OpenLane-V2 benchmark on lane segment. Results for existing methods are from LaneSegNet. "-" denotes the absence of relevant data. We are more focused on $TOP_{lsls}$.

| Method | mAP↑ | $AP_{ls}$ ↑ | $AP_{ped}$ ↑ | v2.0.0 | v2.1.0 |
|--------|------|-------------|--------------|--------|--------|
| | | | | $TOP_{lsls}$ ↑ | $TOP_{lsls}$ ↑ |
| TopoNet [1] | 23.0 | 23.9 | 22.0 | 1.0 | - |
| MapTR [41] | 27.0 | 25.9 | 28.1 | - | - |
| MapTRv2 [42] | 28.5 | 26.6 | 30.4 | - | - |
| LaneSegNet [12] | 32.6 | 32.3 | 32.9 | 8.1 | 25.4 |
| **TopoLogic** | **33.2** | **33.0** | **33.4** | **22.0** | **30.8** |

with LaneSegNet (**33.2** v.s. 32.6). Moreover, there is a significant enhancement in topology reasoning score $TOP_{lsls}$ (**22.0** v.s. 8.1 on v2.0.0, **30.8** v.s. 25.4 on v2.1.0).

## 4.4 Alation Study

We have studied several important components of TopoLogic and conducted ablation experiments on the OpenLane-V2 *subset_A*. In the following text, we employ evaluation metrics from the latest v2.1.0 release for our assessment.

**The design of mapping function.** We have studied the effect of different mapping functions in the transformation from lane geometric distances to lane topology. Table 3 suggests that our designed learnable mapping function performs better at mapping the geometric distances of lane lines to lane topology compared to sigmoid-based, tanh-based, and Gaussian functions. It exhibits the best performance in terms of both the lane topology reasoning score and the centerline score (**29.9** v.s. 28.9 v.s. 28.7 v.s. 27.6 on $DET_l$, and **23.9** v.s. 21.7 v.s. 19.1 v.s. 15.1 on $TOP_{ll}$).

**The approach of lane topology reasoning.** We have investigated the impact of various lane topology computation approaches on lane topology reasoning, specifically using MLP, lane query similarity, and geometric distance. The results in the Table 4 indicate that the integration of a lane topology method, combining both lane geometric distance and lane query similarity calculations, yields optimal results on $TOP_{ll}$ (**23.9** v.s. 20.1 v.s. 12.9 v.s. 10.8) and also performs best in terms of lane line detection scores indicated by $DET_l$ (**29.9** v.s. 28.6 v.s. 28.1 v.s. 27.8). This demonstrates that

Table 3: Ablation study on different mapping functions from lane geometric distance to lane topology on centerline.

| Function | DET$_l$ ↑ | DET$_t$ ↑ | TOP$_{ll}$ ↑ | TOP$_{lt}$ ↑ | OLS↑ |
|---|---|---|---|---|---|
| $f_{tan}$ | 27.6 | 47.2 | 15.1 | 24.8 | 40.9 |
| $f_{sig}$ | 28.7 | 44.1 | 19.1 | 24.1 | 41.4 |
| $f_{gau}$ | 28.9 | 46.8 | 21.7 | 23.2 | 42.6 |
| $\boldsymbol{f}_{ours}$ | **29.9** | **47.2** | **23.9** | **25.4** | **44.1** |

Table 4: Ablation study on different lane topology reasoning approaches on centerline. Ours indicate Similarity+GeoDist.

| Approach | DET$_l$ ↑ | DET$_t$ ↑ | TOP$_{ll}$ ↑ | TOP$_{lt}$ ↑ | OLS↑ |
|---|---|---|---|---|---|
| MLP | 27.8 | 46.8 | 10.8 | 23.9 | 39.1 |
| Similarity | 28.1 | 46.4 | 12.9 | 23.7 | 39.8 |
| GeoDist | 28.6 | 44.1 | 20.1 | 23.1 | 41.4 |
| **Ours** | **29.9** | **47.2** | **23.9** | **25.4** | **44.1** |

Table 5: Ablation study on using MLP to encode lane query computing lane similarity topology. Ours indicate using two indepent MLPs.

| Approach | DET$_l$ ↑ | DET$_t$ ↑ | TOP$_{ll}$ ↑ | TOP$_{lt}$ ↑ | OLS↑ |
|---|---|---|---|---|---|
| No MLP | 25.6 | 46.5 | 18.7 | 20.8 | 40.2 |
| Single MLP | 27.5 | 46.8 | 21.2 | 23.8 | 42.3 |
| **Ours** | **29.9** | **47.2** | **23.9** | **25.4** | **44.1** |

the topology obtained from the fusion of these two methods can also improve the learning of lane centerline through GNN feature enhancement.

**The approach of similarity topology.** We investigated the impact of using MLPs to encode lane queries for computing lane similarity on lane topology reasoning. As Lane similarity topology, its subtlety lies in that it uses two independent MLPs to map the lane query rather than a single MLP, which can decouple a lane into two queries of start and end point, achieving an analogous effect to the geometric distance approach in semantical space. The results presented in Table 5 indicate that encoding lane query with two independent MLPs to obtain features for the starting and ending points of lanes allows for better computation of lane similarity topology. This approach exhibits the best performance in terms of both lane centerline detection score and lane topology reasoning score (**29.9** v.s. 27.5 v.s. 25.6 on DET$_l$, and **23.9** v.s. 21.2 v.s. 18.7 on TOP$_{ll}$).

**The post-processing mode of geometric distance approach.** We have investigated the effectiveness of geometric distance approach as a post-processing module on well-trained model. In Table 6, we conducted experiments by adding a post-processing to the already trained TopoNet, SMERF, and LaneSegNet, respectively. The results indicate that our proposed approach, which calculates lane topology based on lane geometric distances, can be integrated into well-trained models without any additional modifications and can significantly enhance the performance of lane topology reasoning(**22.3** v.s. 10.9 on TopoNet, **26.2** v.s. 15.4 v.s. on SMERF).

## 4.5 Qualitative Analysis

As shown in Figure 5, we presents a qualitative comparison between TopoLogic and TopoNet. Specifically, two traffic scenes are selected for analysis, and the results of lane line detection and topology reasoning are visualized. The first row displays multi-view inputs of realistic scenes, while the second row shows the lane detection results of TopoLogic and TopoNet alongside the ground truth. Notably, TopoLogic demonstrates superior accuracy in lane line detection compared to TopoNet.

**Lane Graph.** The inherent complexity of topology reasoning makes intuitive representation of results challenging. To address this issue, as shown in third row in Figure 5, we construct a lane graph where nodes represent lane lines, and their relative positions in the graph correspond one-to-one with the relative positions of lane lines. This layout enhances the connection between lane line detection results and facilitates subsequent analysis. Additionally, we use directed edges to represent the lane topology, with red indicating error predictions and blue indicating missing predictions. TopoLogic consistently exhibits proficient lane topology reasoning across various intersection scenarios, showcasing significantly enhanced performance in the topology graph compared to TopoNet.

Table 6: Ablation study on incorporating lane geometric distance into post-processing for well-trained model under different task settting (centerline / centerline+SDMap / lane segment).

| Centerline | $\text{TOP}_{ll} \uparrow$ | Centerline+SDMap | $\text{TOP}_{ll} \uparrow$ | Lane Segment | $\text{TOP}_{lsls} \uparrow$ |
|---|---|---|---|---|---|
| TopoNet [1] | 10.9 | SMERF [13] | 15.4 | LaneSegNet [12] | 25.4 |
| **TopoNet+GeoDist** | **22.3** | **SMERF+GeoDist** | **26.2** | **LaneSegNet+GeoDist** | **29.6** |

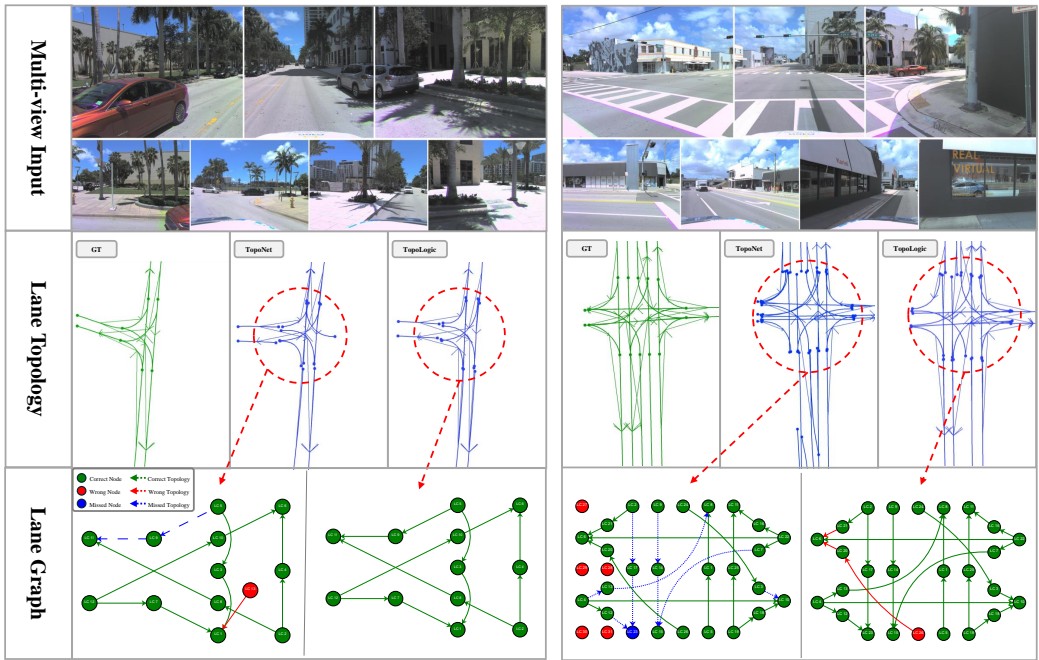

Figure 5: **Qualitative result about lane topology reasoning result of TopoNet and our TopoLogic.** The first row denotes multi-view inputs. The second row denotes lane detection result and lane topology reasoning result. The third row denotes graph form of lane topology reasoning (node indicates lane line, edge indicates lane topology), where green color indicates the right prediction, while red color indicates the error prediction and blue color indicates missing prediction.

## 5 Conclusion

In this paper, we reveal the limitation of using vanilla MLP in lane topology reasoning task and propose TopoLogic, which is the first to employ an interpretable approach for lane topology reasoning. TopoLogic fuses the geometric distance of lane line endpoints mapped through a designed function and the similarity of lane query in a high-dimensional semantic space to reason lane topology. Experiments on the large-scale autonomous driving dataset OpenLane-V2 benchmark demonstrate that TopoLogic significantly outperforms existing methods in topology reasoning in complex scenarios.

**Limitations.** Due to the GNN's role in merely aggregating features from adjacent lanes to enhance the learning of the current lane, our proposed method significantly improves the performance of lane topology but does not substantially elevate lane detection.

**Impact.** Based on the previous sections, it is evident that our proposed method is intended for research purposes. It should not be directly used in or deployed within any actual autonomous driving application. Notably, we cannot provide any guarantee in safety-critical situations.

## 6 Acknowledgements

This work is supported by National Key R&D Program of China (2023YFD2000303) and National Natural Science Foundation of China (62372433).

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
