# OpenReview forum: "TopoLogic: An Interpretable  Pipeline for Lane Topology Reasoning on Driving Scenes"
_NeurIPS.cc/2024/Conference — NeurIPS 2024 poster_

### Official Review · Reviewer_E2Wq · 2024-07-09

**Soundness:** 3
**Presentation:** 3
**Contribution:** 2
**Rating:** 4
**Confidence:** 4

**Summary:**

The paper investigates topological reasoning for lane graph prediction for autonomous driving. The paper focuses on improving the prediction of the topological structure of the lane graph. Therefore, they propose two new mechanisms, the first is a geometric approach that estimates connectivity based on the distance between the end point and starting point of center lines. Secondly, they propose a semantic approach which uses the similarities of lane queries in a high dimensional space. These two approaches stand in contrast to existing works which mainly use MLPs to predict the connectivity of the lanes. They show how these approaches can be interleaved into a lane graph network and how it improves results, especially the topological reasoning results. Even adding only the geometric approach to existing methods improves the topological reasoning results.

**Strengths:**

The paper is well written and easy to follow. The paper and results highlight the importance of geometric cues when predicting conductivity. Or in other words, the fact that the geometric approach works so well shows that there is a lot of potential in current methods. Furthermore, the geometric approach can be integrated into existing networks or combined with other approaches and allows for competitive lane topology prediction.

Both the geometric and semantic prediction branches are clear and make sense and are able to improve results for topological metrics.

**Weaknesses:**

- It is not clear to me why the paper does not compare with TopoMLP, the numbers for Table 2 v1.0.0 can just be taken out of the paper.
- When comparing with TOP_lt TopoMLP outperforms TopoLogic and overall performance is very similar, in this case both methods use the same backbone.
- The paper highlights that geometric cues in predicting the topology of the lane graph are underutilized which is interesting, but I am not fully sure if this is a Neurips paper, since technical contributions are marginal.
- See questions

**Questions:**

- Given that f_ours is just a gaussian function with an adapted decision boundary. I think the fair comparison would be with a hand tuned decision boundary, for example by looking at end point errors of a trained model.
- f_tanh in equation 6 seems to map to [0, 2] and not to [0, 1]
- Equation 10 is not a [0, 1] probability anymore, to remain this property you would need to take a convex combination of the two values. Do you perform some sort of normalization?
- If you say a variable is 1x1 do you mean it is a real number? If so just write \in R
- The batch size of 2 on 8 GPUs seems strange, is it a batch size of 2 per GPU, so an actual batch size of 16?
- How do you handle traffic elements?

- Typos:
- L90: . some methods - not capitalized after full stop
- L204: Both models - three models are listed before
- L236: ),especially
- L259: Table 4 Indicate show

**Limitations:**

Has been discussed

---

> ### Author Rebuttal · Authors · 2024-08-06
>
> **Thanks for your careful reading of our paper. We hope our response and clarifiction can ease some of your concerns and you could reconsider your rating.**
>
> **Q1:**  Why the paper does not compare with TopoMLP?
>
> **A1:** Thank you for your insightful question. I will respond to it in detail. Although both TopoMLP and TopoLogic use ResNet-50 as their backbone, TopoMLP is implemented using PETR, while TopoLogic, TopoNet, SMERF, and LaneSegNet are based on DETR. **Unfortunately, TopoMLP could not be run on the NVIDIA RTX 3090 GPU due to its high memory consumption. As a result, we were unable to perform a comparative analysis with TopoMLP at that time due to equipment limitations.** We apologize for this inconvenience. To address this issue, **we have implemented TopoLogic based on PETR and conducted experiments on Tesla A100 GPU**. The experimental results are provided below, and we will include these results in the final version of the paper.
>
> | Method     | DET$_l$  | DET$_t$  | TOP$_{ll}$ | TOP$_{lt}$ | OLS      |
> | ---------- | -------- | -------- | -------- | -------- | -------- |
> | TopoMLP    | 28.3     | **50.0**     | 19.0     | 23.4     | 42.2     |
> | TopoLogic  | **29.9** | 47.2     | 23.9     | 25.4     | 44.1     |
> | TopoLogic\* | 29.7     | 49.5 | **25.4** | **26.8** | **45.3** |
>
> The experiments were conducted on subset_A, OpenLane-V2 version 2.1.0. In this context, **TopoLogic\* denotes the version of TopoLogic implemented using PETR.**
>
> ----
>
> **Q2:** TopoMLP outperforms TopoLogic on TOP$_{lt}$?
>
> **A2:** As indicated in the table in A1, TopoMLP does not outperform TopoLogic on TOP$\_{lt}$ (25.4 v.s. **26.8**). We implemented TopoLogic using PETR, and it shows improved performance on TOP$\_{lt}$ (**26.8** v.s. 25.4 v.s. 23.4).
>
> ----
>
> **Q3:** Not fully sure if this is a Neurips paper, since technical contributions are marginal.
>
> **A3:** Thanks.  The performance can **significantly improved** (**+11.4** TOP$\_{ll}$ in Table 5) even if it is only used as post-processing, and can be **further improved** (**+13.0** TOP$\_{ll}$ in Table 1) after integrated training. Such an improvement exceeds the improvement of previous works in a large margin, which **in itself** is a valuable contribution to the field. **reasonable analysis** and **effective verification** are our academic contributions, which will be beneficial to promote the development of the field.
>
> ----
>
> **Q4:** Fair comparison about mapping function.
>
> **A4:** Thank you for your insightful feedback. We appreciate your suggestion to ensure a fair comparison by considering a hand-tuned decision boundary. In response to your concern, we conducted additional experiments to further validate the performance of $f_{ours}$ Gaussian function with its adaptive decision boundary.
>
> To determine the decision boundary, we explored two hyperparameters: $\alpha$ and $\lambda$. We evaluated the TOP$\_{ll}$ metrics across various combinations of these hyperparameters. The results are presented in the following table:
>
> | $\alpha$ | $\lambda$ | TOP$\_{ll}$ |      | $\alpha$ | $\lambda$ | TOP$\_{ll}$ |
> | ---------- | --------- | ---------- | ---- | -------- | ----------- | ---------- |
> | 2.0        | 0.23      | 20.4       |      | 1.3      | 0.06        | 16.0       |
> | 1.8        | 0.23      | 21.0       |      | 1.3      | 0.10        | 19.7       |
> | 1.6        | 0.23      | 21.8       |      | 1.3      | 0.14        | 21.6       |
> | 1.4        | 0.23      | 22.5       |      | 1.3      | 0.18        | 22.7       |
> | **1.3**    | **0.23**  | **23.9**   |      | **1.3**  | **0.23**    | **23.9**   |
> | 1.2        | 0.23      | 22.8       |      | 1.3      | 0.26        | 23.1       |
>
> As the table illustrates, **the decision boundary with $\alpha=1.3$ and $\lambda=0.23$ yielded the best performance in terms of the TOP$\_{ll}$ metric.** This finding aligns with our original experimental results and underscores the optimization effectiveness and advantages of the $f_{ours}$ Gaussian function. We hope these additional results address your concerns regarding fair comparison and demonstrate the robustness of our approach.
>
> ----
>
> **Q5:** $f_{tan}$ in equation 6 seems to map to [0, 2] and not to [0, 1].
>
> **A5:** Thank you for your detailed consideration. In fact, since $f$ maps endpoint distances, so $x$ is always greater than or equal to 0 rather than unbounded, ensuring that the range of these functions is always within [0, 1].
>
> ----
>
> **Q6:** Equation 10 is not a [0, 1] probability, and do you perform some sort of normalization?
>
> **A6:** Thank you for your attention. We did not perform normalization. The two coefficients, λ₁ and λ₂, in Equation 10 are learnable and can be trained to range between [0,1]. In other words, λ₁ + λ₂ approaches 1.
>
> ----
>
> **Q7:** If you say a variable is 1x1 do you mean it is a real number? If so just write \in R.
>
> **A7:** Yes, it's not necessary to say 1x1 shape for scalars, and we're going to modify the expression.
>
> ----
>
> **Q8:** The batch size of 2 on 8 GPUs seems strange, is it a batch size of 2 per GPU, so an actual batch size of 16?
>
> **A8:** Your understanding is correct.  To be precise, the batchsize of each GPU is 2, so the total batchsize of 8  GPUs is 16. It's our fault to make it unclear.
>
> ----
>
> **Q9:** How do you handle traffic elements?
>
> **A9:** For the traffic element treatment, we adopt  the same  strategy as baseline method TopoNet, i.e., predict 2D bounding box of traffic elements using DETR, and reason topology between them with lane utilizing vanilla MLP. Since our work focused on lane topology, we rarely mentioned traffic elements, which will be added in the final version.
>
> ----
>
> **Q10:** Typos.
>
> **A10:** Thank you for carefully pointing out these typos. We will diligently correct these issues.
>
>
>
> **We hope this response could help address your concerns, and wish to receive your further feedback soon.**

---

### Official Review · Reviewer_SnEK · 2024-07-09

**Soundness:** 2
**Presentation:** 2
**Contribution:** 2
**Rating:** 4
**Confidence:** 4

**Summary:**

Reasoning about lane topology is becoming more and more important for the autonomous driving community. Current methods mostly focus on improving the perception performance and ignore the reasoning part of the task. The authors raise the importance of the relationship between lanes, namely the lane distance and lane similarity. The lane distance is determined by the geometry distance between the starting and ending points of the lanes. Then, the connectivity is decided based on the result of a learnable threshold function. The lane similarity is calculated based on the lane queries, and a large similarity indicates a large possibility of connectivity. Both lane distance and lane similarity modules serve as plug-in modules to existing networks. Experimentally, both modules have significantly improved the existing methods.

**Strengths:**

- The authors observed that current methods on lane topology reasoning mostly focus on improving perception accuracy but ignore topology reasoning. The proposed modules serve as a good posting-processing technique for topology reasoning.
- The proposed modules are effective and can be placed in existing networks.

**Weaknesses:**

- In the experiment session, the authors provide scores on lane segment prediction. However, the lane segment is different from the lane centerline. Only the lane centerline is discussed in the methodology session.
- It is not stated which split of the dataset is used for experiments.
- The mixture of using different versions of the metric is confusing (vx.0 and vx.1). As stated in the repository of the dataset, this difference leads to different TOP scores. For instance, in Table, metrics in v2.0 and v2.1 are both listed, which is unreasonable.
- The two proposed modules focus on topology reasoning. However, experimentally, the detection scores of the proposed pipeline also improved compared to other methods. This improvement is not discussed.

**Questions:**

- The two proposed modules are plug-in modules on current networks, and the proposed pipeline is trained in an end-to-end manner. However, as these two modules focus on topology reasoning, and the final topology prediction is decided by these two modules, it is curious to know how these two modules benefit existing networks. Namely, only parameters in the proposed modules are trained, and the remaining weights of the existing networks are frozen.
- For the second proposed module, the lane similarity module. Logically, the lane queries usually encode positional information of lanes. It is curious to know why improving the similarity between two connected lanes would improve topology reasoning performance. That is, the only similar thing between two connected lanes is the ending point of one lane and the starting point of another. Two parallel lanes should have a larger similarity than two connected lanes.

---

> ### Author Rebuttal · Authors · 2024-08-06
>
> **Thanks for your careful reading of our paper. We hope our response and clarifiction can ease some of your concerns and you could reconsider your rating.**
>
> **Q1:** Lane segment is not discussed in the methodology session.
>
> **A1:** Thanks for your attention to this detail, which is indeed omitted from the Methods section. However, lane segment is just **another more refined representation of the lane**, and the centerline can still be **easily extracted** from the lane segment, so that our algorithm does not need any special processing for the lane segment.
>
> ----
>
> **Q2:** It is not stated which split of the dataset is used for experiments.
>
> **A2:** Thanks for your mind. Perhaps we should  this information in the caption of tables, even although these information can be found on line 233-234 and 183-185 (SOTA comparison, Table 1 on subset_A/B & Table 2 on subset_A)  and line 248-250 (ablation study, Table 3-5 on subset_A).
>
> ----
>
> **Q3:** The mixture of using different versions of the metric is confusing (vx.0 and vx.1).
>
> **A3:** It is indeed unusual to using two versions of metric, but it is a legacy issue in the lane topology field, as can be seen in the Openlane-V2 official repository ([Update on OpenLane-V2 Metric · Issue #76 · OpenDriveLab/OpenLane-V2 · GitHub](https://github.com/OpenDriveLab/OpenLane-V2/issues/76)). We provide two versions of the metric **for comparison with more methods**. Moreover, it should be noted that our method has a **significant advantage** over each other **regardless of which version of metric**.
>
> ----
>
> **Q4:** Why detection scores of the proposed pipeline also improved compared to other methods?
>
> **A4:** The reason for this can be found in the caption of Figure 2 and Introduction (Line 61-62), and we will add further explanation in the experiment section. Note that the proposed modules serve in **each** layer of the decoder, where geometric distance topology and lane similarity topology are fused into the final lane topology. This lane topology is facilitated  to augment lane learning **by GNN aggregating features from adjacent lanes** in the **next** decoder layer. Thus, Improvement on TOP$\_{ll}$ corresponds to increased DET$_l$（**28.6** to **29.9** (**+1.3**) in Table 1).
>
> ----
>
> **Q5:** How these two modules benefit existing networks?
>
> **A5:**  Thank you for your valuable suggestions. We carried out experiments by freezing the network parameters of TopoNet and incorporating the similarity and geometric distance modules for end-to-end training. The results in the table indicate that the two modules introduced in TopoLogic significantly enhance TopoNet. Additionally, training TopoLogic end-to-end yields better performance compared to freezing a portion of the parameters.
>
> | Method                                | DET$_l$  | DET$_t$  | TOP$\_{ll}$ | TOP$_{lt}$ | OLS      |
> | ------------------------------------- | -------- | -------- | ---------- | ---------- | -------- |
> | TopoNet (baseline)                    | 28.6     | **48.6** | 10.9       | 23.8       | 39.8     |
> | TopoLogic (freeze TopoNet parameters) | 29.2     | 46.9     | 22.8       | 24.8       | 43.4     |
> | TopoLogic                             | **29.9** | 47.2     | **23.9**   | **25.4**   | **44.1** |
>
> ----
>
> **Q6:** Why improving the similarity between two connected lanes would improve topology reasoning performance?
>
> **A6:** Thank you for your valuable question; it is very meaningful. The similarity module proposed in TopoLogic calculates the similarity of **lane endpoints** rather than **the entire lane itself**. As shown in Figure 1, the lane query passes through **two independent MLPs** to encode representations, computing the similarity between **the starting and ending points** of two directed lanes. Additionally, to address your inquiry, we conducted experiments with MLPs as shown in the table. **The "No MLP" configuration does not encode the lane query, calculating the similarity of the entire lane. In contrast, the "Single MLP" configuration computes the similarity of the same endpoint across two lanes.** The table indicates that employing two independent MLPs to encode lane queries for calculating the similarity between the starting and ending points of directed lanes yields the best performance, thereby validating the effectiveness of the TopoLogic design.
>
> | Method               | DET$_l$  | DET$_t$  | TOP$\_{ll}$ | TOP$_{lt}$ | OLS      |
> | -------------------- | -------- | -------- | ---------- | ---------- | -------- |
> | No MLP               | 25.6     | 46.5     | 18.7       | 20.8       | 40.2     |
> | Single MLP           | 27.5     | 46.8     | 21.2       | 23.8       | 42.3     |
> | Two independent MLPs | **29.9** | **47.2** | **23.9**   | **25.4**   | **44.1** |
>
>
>
> **We hope this response could help address your concerns, and wish to receive your further feedback soon.**

---

> > ### Comment · Reviewer_SnEK · 2024-08-08
> >
> > Thank you for your feedback. I still got some question.
> >
> > For Q2, are the scores reported on the validation set or test set?
> >
> > For Q5, the DET_t of the frozen TopoNet drops compared to the baseline version, are there modules which could affect the traffic element detection branch?

---

> ### Author Response · Authors · 2024-08-09
>
> **Thank you very much for your quick response.**
>
> **Q7:** For Q2, are the scores reported on the validation set or test set?
>
> **A7:** We're sorry we didn't mention that detail, and we will add more clarification in the final version. In fact, all the results are reported on the **validation set**, which is **completely consistent** with the protocol of the **all reference paper**, e.g., TopoNet, SMERF, LanSegNet, etc. As for test set, it is only used for the closed-evaluation of the OpenLane-V2's challenge(https://opendrivelab.com/challenge2024/#mapless_driving), which was not available at the time of our research.
>
> ----
>
> **Q8:** For Q5, the DET$_t$ of the frozen TopoNet drops compared to the baseline version, are there modules which could affect the traffic element detection branch?
>
> **A8:** We're sorry to confuse you.  Either similarity module or geometric distance module just involves the **lane**, not **traffic elements**, so DET$_t$ really shouldn't drop. However, in the freezing experiment we loaded the weights of the TopoNet that we reproduced, whose DET$_t$ in itself is lower (**46.9** v.s. **48.6**) than reported in the TopoNet paper. It seems to be **a bug in baseline's official code**. The reproduction about DET$_t$ was always **unstable and lower** than the value reported in the original paper, and this phenomenon can also be observed in the results of other paper (e.g., **44.5** DET$_t$ of TopoNet in Table 1 in paper SMERF$^{[1]}$)
>
> [1] Luo K Z, Weng X, Wang Y, et al. Augmenting lane perception and topology understanding with standard definition navigation maps[J]. arXiv preprint arXiv:2311.04079, 2023.
>
> **If you have any further questions or suggestions about our article, we'd love to discuss our content with you.**

---

### Official Review · Reviewer_FMUS · 2024-07-12

**Soundness:** 3
**Presentation:** 3
**Contribution:** 3
**Rating:** 5
**Confidence:** 3

**Summary:**

The authors claim previous topology reasoning methods typically boost reasoning performance by enhancing the perception of lanes and directly adopt MLP to learn lane topology from lane query. In this work,  the authors propose to make full use of lane geometric distance and lane query similarity for topology reasoning. The proposed method, named TopoLogic, mitigates the impact of endpoint shifts in geometric space, and introduces explicit similarity calculation in semantic space as a complement. The method achieves SOTA results in OpenLane-V2 dataset.

**Strengths:**

1. The idea of paying more attention on geometry and similarity to help topology reasoning makes sense. It makes the reasoning process more interpretable and robust.
2. The performance gain brought by the proposed method is significant. The authors provide adequate ablation experiments to validate the design.
3. Many qualitative results are presented to show the effectiveness of the proposed method. The conection between lane endpoints are obviously improved.

**Weaknesses:**

1. Like an incremental work based on TopoNet. The contribution is limited. The contributed part is only a similarity module for mapping lane line endpoints. But the design is simple and straightforward, not surprising enough.
2. The proposed GeoDist module is a post-processing algorithm. It's like an engineering improvement rather than academic contribution.

**Questions:**

When compared with previous methods (like TopoNet, SMERF), what about the computation overhead and the difference in network architecture?

**Limitations:**

Refer to the Weaknesses part.

---

> ### Author Rebuttal · Authors · 2024-08-06
>
> **Thanks for your careful reading of our paper. We hope our response and clarifiction can ease some of your concerns and you could reconsider your rating.**
>
> **Q1:** Like an incremental work based on TopoNet. The contribution is limited. The contributed part is only a similarity module for mapping lane line endpoints. But the design is simple and straightforward, not surprising enough.
>
> **A1:** Thanks for your comments. Compared to TopoNet, our modification looks like small, but that doesn't mean the contribution is small.
>
> - We clearly point out that TopoNet and other methods ignore the endpoint-connection characteristics between the lanes when topological reasoning, and design **a module based geometric distance** and **a module based on semantic similarity** to **significantly improve** the topological performance (**+13.0** TOP$\_{ll}$  in Table 1) in the case of **reducing the number of parameters and the amount of computation**. Therefore, the contributed part is not limited in similarity module.
> - As similarity module, its subtlety lies in that it uses **two independent MLPs** to map the lane query **rather than a single MLP**, which can **decouple a lane into two queries of start and end point**, achieving an **analogous** effect to the (endpoint) geometric distance module **in semantical space**. This opinion can be verified by the following experiments, so it is not simple and straightforward.
>
> |                      | DET$_l$ | DET$_t$ | TOP$\_{ll}$ | TOP$_{lt}$ | OLS  |
> | -------------------- | ------- | ------- | ---------- | ---------- | ---- |
> | No MLP               | 25.6    | 45.9    | 18.7       | 20.8       | 40.0 |
> | Single MLP           | 27.5    | 46.8    | 21.2       | 23.8       | 42.3 |
> | Two independent MLPs |**29.9**    | **47.2**    | **23.9**       | **25.4**       | **44.1** |
>
> ----
>
> **Q2:** The proposed GeoDist module is a post-processing algorithm. It's like an engineering improvement rather than academic contribution.
>
> **A2:** Thanks for your comments, but we don't think it's just an engineering improvement.
>
> - The GeoDist module is not just used as post-processing after training. This module can be used (in conjunction with Similarity module) at each layer of the decoder, so that the lane topology is enhanced geometrically and semantically at each layer and  refined layer by layer. In addition, lane topology  in  layer is also facilitated  by GNN to augment lane learning  in   layer through aggregating features from adjacent lanes. Using GeoDist module in an integrated training manner provides more performance gains (**+13.0** TOP$\_{ll}$  in Table 1) than just post-processing (**+11.4** TOP$\_{ll}$  in Table 5).
>
> - This improvement is not an accident attempt,  but comes from our **accurate and highly interpretable understanding**  of **the endpoint-connection characteristics** of the lane topology, which makes it apart from simple engineering improvements.
>
>   All in one, **reasonable analysis** and **effective verification** are our academic contributions.
>
> ----
>
> **Q3:** When compared with previous methods (like TopoNet, SMERF), what about the computation overhead and the difference in network architecture?
>
> **A3:** Thank you for your valuable question. The key distinction between TopoLogic and both TopoNet and SMERF is that **TopoLogic introduces similarity module and geometric distance module for lane topology reasoning, whereas TopoNet and SMERF only use MLP for lane topology reasoning.** Here, we provide some computational benchmark as follows, and we will include the experiment result in the final version of the paper.
>
> |                         | SDMap | #PARAMS | FLOPS  | FPS  |
> | ----------------------- | ----- | ------- | ------ | ---- |
> | TopoNet                 | false | 38.6M   | 712.1G | 10.5 |
> | TopoLogic(TopoNet ver.) | false | 37.8M   | 665.0G | 10.8 |
> | SMERF                   | true  | 39.4M   | 720.1G | 15.3 |
> | TopoLogic(SMERF ver.)   | true  | 38.7M   | 678.2G | 15.8 |
>
>
>
> **We hope this response could help address your concerns, and wish to receive your further feedback soon.**

---

> > ### Comment · Reviewer_FMUS · 2024-08-12
> > **Further questions**
> >
> > Thanks for the authors' feekback. My further questions are as follows:
> > Why does TopoNet-based TopoLogic and SMERF-based TopoLogic run faster and have less params and flops compared with original TopoNet and SMERF? Is the post-processing step (GeoDist module) included？

---

> > > ### Author Response · Authors · 2024-08-12
> > >
> > > **Thank you for your insightful question.**
> > >
> > > **Q4**: Why does TopoNet-based TopoLogic and SMERF-based TopoLogic run faster and
> > >  have less params and flops compared with original TopoNet and SMERF?
> > >
> > > **A4**: TopoNet and SMERF both compute lane topology directly using MLP. In contrast, TopoLogic calculates lane topology through a geometric distance module and a similarity module. This distinction highlights the structural differences between TopoLogic and both TopoNet and SMERF.
> > >
> > > In the TopoNet and SMERF, two MLPs are used to generate $Q_{emb}  \in \mathbb{R}^{N \times C}$, where $N$ is the number of query. Subsequently, $Q_{emb}$ are repeated to $Q_{emb}^{'} \in  \mathbb{R}^{N \times N \times C}$. **They  are then concatenated and processed through $\operatorname{MLP_3}$ to compute the lane topology**, as described by the following Equations:
> > > $$
> > > Q_{emb_1}, Q_{emb_2}=\operatorname{MLP_1}(Q_l^i), \operatorname{MLP_2}(Q_l^i) \in \mathbb{R}^{N \times C}
> > > $$
> > > $$
> > > Q_{emb_1}^{'},Q_{emb_2}^{'}=\operatorname{Repeat}(Q_{emb_1}),\operatorname{Repeat}(Q_{emb_2}) \in \mathbb{R}^{N \times N \times C}
> > > $$
> > > $$
> > > \operatorname{G_{sim}}=\operatorname{MLP_3}(\operatorname{Concat}(Q_{emb_1}^{'}, Q_{emb_2}^{'})) \in \mathbb{R}^{N \times N}
> > > $$
> > >
> > > For TopoLogic, we calculates lane topology through a geometric distance module and a similarity module. In similarity module, We directly compute lane topology via matrix multiplation between $Q_{emb_1}$ and the transposition of $Q_{emb_2}$, followed by applying a sigmoid activation function, as detailed in our paper:
> > >
> > >
> > > $$
> > > Q_{emb_1}, Q_{emb_2}=\operatorname{MLP_1}(Q_l^i), \operatorname{MLP_2}(Q_l^i) \in \mathbb{R}^{N \times C}
> > > $$
> > > $$
> > > \operatorname{S}=\operatorname{matmul}(Q_{emb_1},\operatorname{transpose}(Q_{emb_2})) \in \mathbb{R}^{N \times N}
> > > $$
> > > $$
> > > \operatorname{G_{sim}}=\operatorname{sigmoid}(\operatorname{S}) \in \mathbb{R}^{N \times N}\\
> > > $$
> > >
> > > In geometric distance module, as illustrated in Equation (4) of our paper, we utilize only two parameters, $\alpha$ and $\lambda$. Additionally, the merge process involves two parameters, $\lambda_1$ and $\lambda_2$, as shown in Equation (10) of our paper. All these parameters are scalar values,  i.e., $\alpha,\lambda, \lambda_1, \lambda_2\in\mathbb{R}$.
> > >
> > > To summarize, while TopoNet and SMERF models involve **three MLPs** with associated parameters, TopoLogic utilize only **two MLPs** in the similarity module, two parameters $\alpha$, $\lambda$ in geometric distance m odule, and $\lambda_1$ and $\lambda_2$ in merge process. **In TopoLogic, the scalar parameters in the geometric distance module are significantly fewer in number compared to the MLP$_3$ parameters in TopoNet and SMERF.**  Additionally, the MLP$_3$ results in a higher computational complexity. Therefore, TopoLogic has a smaller number of parameters (#PARAMS) and lower computational complexity (FLOPS), leading to faster speeds (FPS) compared to TopoNet and SMERF.
> > >
> > > **We hope this explanation addresses your concern, thank you again for your valuable feedback.**

---

### Official Review · Reviewer_8GTK · 2024-07-16

**Soundness:** 3
**Presentation:** 3
**Contribution:** 2
**Rating:** 5
**Confidence:** 4

**Summary:**

This paper proposes an interpretable method for lane topology reasoning based on lane geometric distance and lane query similarity. The authors reveal that the lane topology is easily disturbed by the endpoint shifts. Based on this, the proposed post-processing module improves the robustness and performance of lane topology reasoning, which can be plugged into other methods without re-training. And it also alleviates the influence of inaccuracies in lane detection on topology reasoning. Extensive experiments on OpenLane-V2 demonstrate its state-of-the-art performance.

**Strengths:**

- The authors reveal the phenomenon that topology reasoning is highly susceptible to lane line endpoint shifts, providing a solid foundation and motivation for subsequent research.
- The proposed post-processing module improves the robustness and performance of lane topology reasoning, which can be plugged into other methods without retraining.
- The entire paper is easy to understand. The lane graph in Figure 5 shows very clearly the topological relationships between lanes and the differences from previous methods, which should be followed by other subsequent work.

**Weaknesses:**

- The proposed method is relatively simple and involves only post-processing modules, which at the same time leads to limited contributions and incremental improvements.

- Considering that TopoNet is just a simple baseline model, the boost in topological reasoning is not surprising. Moreover, as can be seen in Table 1, the improvement of $TOP_{lt}$ is very small and $DET_{t}$ even decreases, which needs further analysis and discussion.

**Questions:**

- As can be seen in Figure 3, compared to other mapping functions, the final result of the learnable Gaussian mapping function has a much higher threshold. Therefore, compared to the present comparison experiments ($f_{gau}, f_{sig}, f_{tan}$), it may be better to manually set the hyperparameters for a series of gradients to observe its effect on the final performance. And I'm wondering why $f_{gau}, f_{sig}, f_{tan}$ are very close but differ dramatically on $TOP_{ll}$ in Table 3.

- According to L146-149, the motivation of Lane Similarity Topology is to alleviate the influence of inaccuracies in lane detection on topology reasoning. However, the proposed method as a post-processing step does not significantly change the lane line detection results. In addition, the comparison results of the Lane Similarity Topology module should be added to Figure 4 to increase its persuasiveness.

There are many typos in the paper, which should be further checked and polished.
For example:
L6: "are prone to" -> "is prone to"
L12: "our methods provides" -> "our methods provide"
L17: "boost" -> "boosting"
Figure 1 caption: "reasoing"

**Limitations:**

This paper is limited in the post-processing for topology reasoning.

---

> ### Author Rebuttal · Authors · 2024-08-06
>
> **Thanks for your careful reading of our paper. We hope our response and clarifiction can ease some of your concerns.**
>
> **Q1:** The proposed method is relatively simple and involves only post-processing modules, which at the same time leads to limited contributions and incremental improvements.
>
> **A1:** Thanks for your comments.
>
> - The proposed method, TopoLogic, is an end-to-end trained model with two key modules (i.e., Similarity & GeoDist) instead of just one simple post-processing. **The modules are plugged at each layer of the decoder, so that the lane topology is enhanced geometrically and semantically at each layer and  refined layer by layer.**  Once trained, the GeoDist module can be utilized as a plug-and-play post-processing component for any pre-trained lane topology reasoning model to enhance TOP$\_{ll}$.
>
> - Although the model is relatively simple, the performance improvement is significant (**10.9** to **23.9** (**+13.0**) on TOP$\_{ll}$ in Table 1) after end-to-end training. **The improvement in itself** is a valuable contribution to the field, and its **theoretical basis** (endpoint-connection characteristics of the lane topology) is another contribution to the field.
>
> ----
>
> **Q2.1:** Considering that TopoNet is just a simple baseline model, the boost in topological reasoning is not surprising.
>
> **A2.1:** Thank you for your feedback. We are to blame for the ambiguity, and we will add more clarification in the final version. **TopoNet is the official recommended baseline evaluated on OpenLane-V2 benchmark, and recently many new model (e.g., SMERF, LaneSegNet) are actually based on TopoNet, so we claim vaguely that our TopoLogic is based on TopoNet.** We conducted experiments with TopoNet, SMERF, and LaneSegNet. As demonstrated in Table 1 and Table 2, TopoLogic consistently achieves superior performance compared to these approaches.
>
> **Q2.2:** The improvement of is very small and even decreases, which needs further analysis and discussion.
>
> **A2.2:** Thanks for your comments, and we will add more clarification in the final version.
>
> - It is **reasonable** that the improvement on TOP$_{lt}$ is smaller the improvement on TOP$\_{ll}$, as our method **mainly affects the lane-lane topology** and **does not deal with traffic elements specifically**.
>
> -  For the same reason, DET$_t$ has not improved. As for decrease sometimes, it seems to be **a bug in baseline's official code**. When we reproduced the DET$_t$ of baseline, it was always **unstable and lower** than the value reported in the original paper. We also found that the DET$_t$ 's reproduced result in other papers (e.g. SMERF) were lower.
>
> ----
>
> **Q3.1:** It may be better to manually set the hyperparameters for a series of gradients to observe its effect on the final performance.
>
> **A3.1:**  Thank you for your feedback. We tested a series of manually set $\alpha$  and $\lambda$ values to observe their effects on performance. As is shown in following table, automatically learned hyperparameters $\alpha=1.3, \beta=0.23$ are actually superior to other manually set value, which frees us from the hassle of tuning parameters.
>
> | $\alpha$ | $\lambda$ | TOP$\_{ll}$ |      | $\alpha$ | $\lambda$ | TOP$\_{ll}$ |
> | -------- | --------- | ----------- | ---- | -------- | --------- | ----------- |
> | 2.0      | 0.23      | 20.4        |      | 1.3      | 0.06      | 16.0        |
> | 1.8      | 0.23      | 21.0        |      | 1.3      | 0.10      | 19.7        |
> | 1.6      | 0.23      | 21.8        |      | 1.3      | 0.14      | 21.6        |
> | 1.4      | 0.23      | 22.5        |      | 1.3      | 0.18      | 22.7        |
> | **1.3**  | **0.23**  | **23.9**    |      | **1.3**  | **0.23**  | **23.9**    |
> | 1.2      | 0.23      | 22.8        |      | 1.3      | 0.26      | 23.1        |
>
> ----
>
> **Q3.2:** Why $𝑓_{𝑔𝑎𝑢}$,$𝑓_{𝑠𝑖𝑔}$,$𝑓_{𝑡𝑎𝑛}$ are very close but differ dramatically on TOP$_{𝑙𝑙}$ in Table 3?
>
> **A3.2:** Thank you for your thoughtful question. We believe the counterintuitive phenomenon is primarily due to **the sensitivity of these functions to the threshold values**. To investigate this hypothesis, we conducted additional analyses by plotting Gaussian function curves with varying $\alpha$ and $\lambda$ parameters, as illustrated in Figure 1 and 2 of the Rebuttal PDF. **These figures and table in A3.1 show that while the threshold values do not exhibit significant changes, the TOP$\_{ll}$ scores vary substantially**, which is another reason why we adopt learnable mapping functions in TopoLogic.
>
> ----
> **Q4.1:** The motivation of Lane Similarity Topology is to alleviate the influence of inaccuracies in lane detection on topology reasoning. However, the proposed method as a post-processing step does not significantly change the lane line detection results.
>
> **A4.1:** Nice question. We are to blame for the ambiguity, and we will add more clarification in the final version. Our motivation is indeed to alleviate the influence of inaccuracies lane detection on topological reasoning, however, our strategy is **not to directly improve lane detection**, but to **robustly improve lane topological reasoning** through the geometric and semantic information of the lane **in the case of inaccurate detection**, which is a brand new perspective  first proposed by us for the topological reasoning task .
>
> ----
>
> **Q4.2:** In addition, the comparison results of the Lane Similarity Topology module should be added to Figure 4 to increase its persuasiveness.
>
> **A4.2:** Thanks for your valuable suggestions, and we will add the module's visualized results to Figure 4. Since OpenReview's reply cannot directly attach pictures, please go to Rebuttal PDF Figure 3 to view.
>
> ----
>
> **Q5:** Typos
>
> **A5:** Thank you for carefully pointing out these typos. We will diligently correct these issues.
>
> **We hope this response could help address your concerns, and wish to receive your further feedback soon.**

---

> > ### Comment · Reviewer_8GTK · 2024-08-12
> >
> > Thanks for the authors' positive feedback.
> >
> > It seems that the results of $TOP_{ll}$ are very sensitive to the hyperparameters. And as shown in Figure 3 of the global rebuttal, the introduction of the similarity module removes many predictions of topology. But both results of (b) and (c) are far away from GT. I think that the existing metrics may not reflect the results well, and the filtering could lead to large variations. In addition, the authors are not able to clearly explain the results of the $DET$. The above makes the quantitative experiment less convincing.
> >
> > Overall, I still have concerns about incremental improvements and contributions, which are also mentioned by other reviewers.

---

> > > ### Author Response · Authors · 2024-08-12
> > >
> > > **We are very grateful that you have carefully reviewed our rebuttal response.**
> > >
> > > Although there are deficiencies in the details, we still believe that this work has sufficient contributions to the field: **we have significantly improved the core metrics (TOP$_{ll}$ & OLS) of the emerging task of topology reasoning on autonomous driving with a easily understandable method.**
> > >
> > > Regarding your further questions, we will answer them one by one below. We hope you can reconsider the rating of our article.
> > >
> > > ----
> > >
> > > **Q6.1:** It seems that the results of TOP$_{ll}$ are very sensitive to the hyperparameters.
> > >
> > > **A6.1:** As you said, TOP$_{ll}$ is sensitive to hyperparameters, but **its actual negative effects are tiny**.
> > >
> > > - On the one hand, the learnable hyperparameters we used can effectively **self-adapt** to the data distribution and **avoid** the tedious manual parameter tuning.
> > > - On the other hand, the results under **all** hyperparameter configurations in our experiments are actually always **significantly higher** than the baseline (**16.0~23.9** v.s. 10.9 on Q3.1, **15.1~23.9** v.s. 10.9 on Table 3 in the paper), which further implies the effectiveness of our methods.
> > >
> > > ----
> > >
> > > **Q6.2:** As shown in Figure 3 of the global rebuttal, the introduction of the similarity module removes many predictions of topology. But both results of (b) and (c) are far away from GT. I think that the existing metrics may not reflect the results well, and the filtering could lead to large variations.
> > >
> > > **A6.2:** We are very sorry that our **improper presentation** in Figure 3 gave you the misunderstanding.
> > >
> > > - To avoid the **visual confusion** caused by excessive topological connections, during visualization in Figure 4 of the main paper and its extended version Figure 3(b,c) of the Rebuttal PDF , only **the right lanes** (blue lines), **the incorrect lanes** (yellow lines) and **the incorrect topological connections** (red arrows) are drawn (as is description in the caption of Figure 4 of the main paper), while **the correct topological connections** are **omitted**, which is why the results of (b,c) look away from GT.
> > >
> > > - You can find that **all** the arrows in the figure are red (wrong), but it is **obviously impossible** that all connections are wrong (red), so please believe that we **deliberately visualized it this way** (although it looks stupid now), rather than **the result itself being bad**. In this perspective,  what the figure shows is that the similarity module can further reduce **false predictions**, rather than **all predictions**.
> > > - Thank you very much for pointing out this issue. We will definitely add the correct predictions in the figure in the final version (using other color, e.g. green), but unfortunately, we **cannot upload** any pictures again during the discussion stage limited by Open Review system. If you still have doubts about the visualization results, you can also pay attention to **Figure 5** in the main paper again. We believe both the **Lane Topology** and the **Lane Graph** in it can definitely make people clearly understand the effect of our method.
> > >
> > > ----
> > >
> > > **Q6.3:** The authors are not able to clearly explain the results of the DET.
> > >
> > > **A6.3:** It is indeed hard to provide a more clear explanation in a short time, however, compared with the little drop on DET$\_{t}$(**-1.4**),  the **improvement in lane & topology** (**+13.0** TOP$_{ll}$, **+4.3** OLS, **+1.3** DET$_l$ [Topologic v.s. TopoNet]) is **more significant**. Besides, the latter is our **core research motivation and improvement goal**. Objectively speaking, compared with the **reproduced TopoNet**, our method is not inferior in terms of DET$\_t$ (**47.2** v.s. 46.9). It means that the issue lies in the reproduction of TopoNet (**46.9** v.s. 48.6) rather than our TopoLogic, which has also puzzled other researchers (e.g., the reproduction of TopoNet has lower DET$\_{t}$ (**44.5** v.s. 48.6) in Table 1 of SMERF paper). If there is still a chance, we will definitely explore this issue in depth and fix it.
> > >
> > > ----
> > >
> > > **If you have any further questions or suggestions about our article, we'd love to discuss our content with you.**

---

### Author Rebuttal · Authors · 2024-08-06

We sincerely thank all reviewers for their valuable time and comments. In order to better response for the questions, we have attached a Rebuttal PDF that includes some figures related to Reviewer 8GTK's questions. We hope that following responses could address reviewers’ concerns.

---

### Author Response · Authors · 2024-08-14
**Authors' Response to All Reviews**

Thanks for the reviewers' careful reading of our paper, and giving valuable suggestions for us to improve the paper. We hope our response and clarifiction can ease some of your concern. If you have any further questions, we are more than happy to discuss with you if time permits.

If you are satisfied after reading our response, we sincerely hope that you could increase the rating of our paper. Thank you.

---

### Decision · Program_Chairs · 2024-09-25

**Decision:**

Accept (poster)

**Comment:**

The paper proposes TopoLogic, an interpretable framework for topology reasoning in autonomous driving. It introduces several modifications to mitigate several issues in existing approaches. e.g. explicit similarity calculation. It brings significant improvement over previous SOTA on OpenLane-v2 benchmark.

Authors have carefully addressed reviewers comments and provide more details. There are two positive reviews and two negative ones. The negative comments are mainly focused on the lack of implementaion details / explaination on the experimental settings / typos / etc. Since it falls into the category of borderline paper, AC steps in, read the paper carefully and rebuttal feedbacks.

There are concerns on the technical contribution as pointed out by reviewers. In AC's view, the field of lane detection is **slowly** evolving over the years. There are not too many research on topology reasoning, which is a critical issue for the next-level of lane detection. Given the importance of the topic, great improvement over previous SOTA and detailed feedback on the rebuttal, AC would champion this work for this time. Please incorporate all the concerns, release the code and contribute to the community.